# *Brassica* Bioactives Could Ameliorate the Chronic Inflammatory Condition of Endometriosis

**DOI:** 10.3390/ijms21249397

**Published:** 2020-12-10

**Authors:** Paula García-Ibañez, Lucía Yepes-Molina, Antonio J. Ruiz-Alcaraz, María Martínez-Esparza, Diego A. Moreno, Micaela Carvajal, Pilar García-Peñarrubia

**Affiliations:** 1Aquaporins Group, Centro de Edafologia y Biologia Aplicada del Segura, CEBAS-CSIC, Campus Universitario de Espinardo-25, E-30100 Murcia, Spain; pgibanez@cebas.csic.es (P.G.-I.); lyepes@cebas.csic.es (L.Y.-M.); mcarvaja@cebas.csic.es (M.C.); 2Department of Biochemistry, Molecular Biology B and Immunology, School of Medicine, University of Murcia Regional Campus of International Excellence “Campus Mare Nostrum”, 30100 Murcia, Spain; maria@um.es (M.M.-E.); pigarcia@um.es (P.G.-P.); 3Biomedical Research Institute of Murcia (IMIB), 30120 Murcia, Spain; 4Phytochemistry and Healthy Foods Lab, Department of Food Science Technology, Centro de Edafologia y Biologia Aplicada del Segura, CEBAS-CSIC, Campus de Espinardo-25, E-30100 Murcia, Spain

**Keywords:** aquaporins, *Brassica*, isothiocyanates, indoles, endometriosis, inflammation

## Abstract

Endometriosis is a chronic, inflammatory, hormone-dependent disease characterized by histological lesions produced by the presence of endometrial tissue outside the uterine cavity. Despite the fact that an estimated 176 million women are affected worldwide by this gynecological disorder, risk factors that cause endometriosis have not been properly defined and current treatments are not efficient. Although the interaction between diet and human health has been the focus of many studies, little information about the correlation of foods and their bioactive derivates with endometriosis is available. In this framework, *Brassica* crops have emerged as potential candidates for ameliorating the chronic inflammatory condition of endometriosis, due to their abundant content of health-promoting compounds such as glucosinolates and their hydrolysis products, isothiocyanates. Several inflammation-related signaling pathways have been included among the known targets of isothiocyanates, but those involving aquaporin water channels have an important role in endometriosis. Therefore, the aim of this review is to highlight the promising effects of the phytochemicals present in *Brassica* spp. as major candidates for inclusion in a dietary approach aiming to improve the inflammatory condition of women affected with endometriosis. This review points out the potential roles of glucosinolates and isothiocyanates from *Brassicas* as anti-inflammatory compounds, which might contribute to a reduction in endometriosis symptoms. In view of these promising results, further investigation of the effect of glucosinolates on chronic inflammatory diseases, either as diet coadjuvants or as therapeutic molecules, should be performed. In addition, we highlight the involvement of aquaporins in the maintenance of immune homeostasis. In brief, glucosinolates and the modulation of cellular water by aquaporins could shed light on new approaches to improve the quality of life for women with endometriosis.

## 1. Introduction

Endometriosis is a frequently diagnosed, incurable, hormone-dependent gynecological disorder characterized by a chronic inflammatory profile and histological lesions generated by the abnormal growth of endometrial-like tissue outside the uterine cavity. These lesions mainly occur within the peritoneal cavity and are engrafted in different locations such as the peritoneum wall, ovaries, colon, and bladder, although it can also develop in distant organs, such as the liver, lung, and brain, among others [1,2]. Endometriosis may appear in diverse forms and locations, and thus it can be classified as peritoneal endometriosis; ovarian endometriosis; deep infiltrating endometriosis (DIE); or extragenital endometriosis [1,2].

It is estimated that approximately 176 million women worldwide are affected by endometriosis, which represents about 10% of women of reproductive age [2]. However, it is suspected that these data could underestimate the real number of affected women, since many of them are accidentally diagnosed after going into surgery for the treatment of other pathologies [1,2]. Women affected by endometriosis may present severe symptoms, which include, among others, chronic pelvic pain, dysmenorrhea, and dyspareunia [1,2,3,4]. In this context, 2–4% of women of reproductive age may suffer from sexual dysfunction due to the painful symptoms produced during intercourse [5]. Furthermore, pelvic pain can be incapacitating, especially in cases of DIE [6,7]. Endometriosis causes infertility in approximately 30% of affected women, likely because of the endometriotic lesion scars present in the reproductive organs [1,2,8]. Furthermore, it has been demonstrated that endometriosis is a disease with a high impact on health care costs, being comparable with other chronic diseases such as diabetes [9].

On the other hand, knowledge of the direct influence of diet on human health has grown quickly in recent years [10,11]. However, to date, no conclusive results relating a diet rich in vegetables with endometriosis development or an improvement in its symptoms have been found [12]. In this regard, Parazzini et al. [13] performed a case-control study with 504 women suffering from endometriosis. Food frequency questionnaires revealed that women with a higher intake of vegetables showed a lower endometriosis risk. Nevertheless, another case-control study based on dietary questionnaires found no association between a higher intake of vegetables and a decrease in endometriosis risk [14]. A more recent study showed that women who consumed more than one serving of cruciferous vegetables per day had a 13% higher risk of developing endometriosis compared with those who ingested them once a week or less [15]. However, these data came from retrospective studies based on self-reported questionnaires, while there is little or no information on the effect of a diet rich in *Brassica* spp. for women affected with endometriosis and even less information about the effect of the bioactive compounds present in cruciferous foods.

Therefore, in this review, we aim to point out the promising effect of the phytochemicals present in *Brassica* vegetables as key components of a diet intended to ameliorate the inflammatory condition of endometriosis patients. We bring together the connections between endometriosis and the scarce available information on the mechanistic effects of the biocomponents present in *Brassica* (glucosinolates and their conjugates: Isothiocyanates and thiols) in this pathology, also taking into account the role of aquaporins.

## 2. Etiology of Endometriosis

Endometriosis is considered a benign disease as it is not caused by transformed malignant cells. Although the mechanisms responsible for the origin and further progression of endometriosis have not been established, it has been proposed that endometriosis may be caused by an increased rate of cell proliferation and/or a decreased apoptotic rate. In this regard, the balance between cell proliferation and apoptosis is altered in the eutopic endometrium of endometriosis patients [16]. Additionally, it has been related to an elevated risk of developing ovarian cancer and, in fact, it shares common characteristics with neoplasia, including hyper-proliferative activity and cell growth, invasion of adjacent tissues, defective apoptotic ability, neoangiogenesis, and sustained local inflammatory responses [17,18,19,20,21].

The etiology of endometriosis is still unknown. The classical explanation for the development of endometriosis involves the escape of endometrial tissue fragments from the uterus into the peritoneal cavity by a retrograde menstruation mechanism, which allows the adhesion and growth of this tissue inside the abdominal cavity [22]. However, while retrograde menstruation is a common process, only 10% of women have endometriosis; therefore, multiple factors must be concomitantly acting to cause the disease. Recently, it has become widely accepted that the actual origin of endometriosis must have multiple causes. In this regard, different individual genetic and environmental predisposition factors, together with a deregulated and inefficient immune system, would allow the attachment and growth of endometrial ectopic tissue [23], which plays a key role in the onset and further progression of the disease. Other postulated hypotheses include (a) the occurrence of coelomic metaplasia, i.e., the peritoneal coelomic epithelium differentiates into endometrial-like cells; (b) neonatal spread of somatic stem cells and peritoneal implantation that remains latent until menarche; and (c) abnormal embryonic Müllerian ducts remains (from which the uterus, fallopian tubes, and upper vagina, develop) [23,24]. Furthermore, epidemiological and experimental evidence suggests that intrauterine exposure to high levels of endocrine-disrupting chemicals during the prenatal stage may play a role in the physiopathology of endometriosis in adulthood [25,26,27,28]. However, these theories have not conclusively demonstrated a unique causal association for all cases of endometriosis.

Furthermore, many epidemiological studies have explored the potential association of endometriosis with multiple risk factors. As recently reviewed by Parazzini et al., 2017 [13], Parasar et al., 2017 [29] and, Shafrir et al., 2018 [30], early age at menarche, taller height, lower adiposity, and low body mass index (BMI) [31,32], as well as infertility [33] have been associated with a higher risk of developing endometriosis, while a higher parity and higher BMI are inversely associated with endometriosis. Additionally, high exposure to environmental endocrine disrupting compounds (EDCs), such as phthalates, bisphenols, dioxins, polychlorinated biphenyls (PCBs), pesticides, and many others contaminants, has been associated with a higher endometriosis risk [34,35,36,37]. On the other hand, inconsistent results have been reported from studies associating endometriosis with several modifiable lifestyle risks factors, including smoking and frequent exercise [38,39,40], as well as caffeine or alcohol intake and level of education [29,41].

Current treatments are based on the estrogen dependence of this pathology [42,43]. However, those hormone-based treatments have negative effects on the associated infertility problem and lead to the induction of disorders similar to menopause, including the loss of bone density. All effects associated with this therapy, together with its low efficiency in many patients, have resulted in the administration of the drugs being stopped, allowing the progression of endometriotic lesions and adherences outside the uterus and increasing the ratio of repeated surgical invasive procedures [1,2,44]. Therefore, there is a great need to identify new molecular targets and potential therapies to facilitate the development of non-invasive techniques and/or to allow better treatment of this pathology.

## 3. Inflammatory Background of Endometriosis

The most characteristic histological features of endometriosis lesions are local inflammation and peritoneal fibrosis. The invasive capacity and resistance to apoptosis differ between the endometrium of women with and without endometriosis, as well as between ectopic and eutopic endometrium of patients with endometriosis. This suggests that the ectopic inflammatory environment plays an important role in the development of endometriotic lesions [1,2,24,45,46,47,48]. Furthermore, whilst retrograde migrated endometrial cells in healthy women are cleared by peritoneal immune cells, ectopic endometrial tissue in endometriosis patients escapes immune surveillance, survives, and is implanted in the peritoneum and other organs.

Macrophages are crucial cells that are involved not only in the immune response induced by infected, injured, or tumor stimuli but also in the promotion of the appropriate regulatory response necessary to restore tissue homeostasis [49]. Macrophages are the prevalent immune cell type in the peritoneal fluid of women with endometriosis and, consequently, they must play an important role in its pathophysiology [50,51]. There is accumulated evidence of an altered and dysfunctional activity of macrophages isolated from the ectopic endometrial tissue located in the peritoneal cavity [50,52]. In this regard, an M2 polarization profile, which corresponds to alternative activation of macrophages towards a tissue-reparative phenotype, would permit the survival, growth, and neovascularization of the ectopic endometriotic tissue [50,51,52,53,54,55,56]. Additionally, other studies have shown a classical M1 polarization profile of macrophage activation in the eutopic endometrium of women with endometriosis in comparison with corresponding healthy controls [57]. The knowledge of molecular mechanisms involved in the activation and polarization of macrophages is crucial to understand the pathophysiology of endometriosis and to allow the design of new therapies. In this regard, it is of great importance to use a proper method to obtain human peritoneal macrophages [58]. On the other hand, dysfunctional natural killer (NK) cells [59] that express a higher number of KIR (killing inhibitory receptors) [60] and a lower level of KAR (killing activating receptors), could play a role in the unsuitable activation and polarization of macrophages by inhibiting their phagocytic capacity and the expression of scavenger receptors [61]. Furthermore, these NK cells induce the activation of T regulatory (Treg) lymphocytes, whose inhibitory activity would allow endometrial cells to escape from local immune surveillance [62,63,64,65,66]. Additionally, it has been described that the number of peritoneal Myeloid-Derived Suppressor Cells (MDSCs) rapidly increases in the presence of endometrial tissue, thus contributing to the inhibited immune response [48,67]. Recently Gou et al. [68] showed that ERβ modulates the production of CCL2 (C-C motif chemokine ligand 2, also known as monocyte chemoattractant protein (MCP)-1) via NF-κB (nuclear factor kappa-light-chain enhancer of activated B cells) signaling in endometriotic stromal cells, recruiting macrophages to ectopic lesions and, thus, promoting the pathogenesis.

Additionally, multiple inflammatory mediators are secreted into the endometriotic milieu by several immune cells [69,70]. These mediators include interleukin (IL)-1β (induces the expression of cyclooxygenase 2 (COX-2) and vascular endothelial growth factor (VEGF), increasing angiogenesis) [71,72]; IL-6 (inhibits NK cell-mediated cytotoxicity) [73]; IL-8, which induces migration and increases the survival, proliferation, and angiogenesis of migrated endometrium [74,75]; transforming growth factor (TGF)-β, which is involved in post-surgical tissue adherence [62]; IL-27, which promotes the development of endometriotic lesions by inducing the secretion of IL-10 by T helper (T_H_) 17 cells [76]; IL-32, which increases cell viability and invasiveness [77]; IL-33, which exacerbates endometriotic lesions by polarizing peritoneal macrophages to M2 subtype secreting IL-1β [55]; prostaglandin E2 (PGE_2_), which, together with IL-4, induces the synthesis of estrogens, increases the production of fibroblast growth factor (FGF)-9, and inhibits the rate of apoptosis [78]; macrophage inhibitory factor (MIF), which induces angiogenesis and endometriotic cell expression of VEGF, IL-8, and MCP-1 (CCL2) [79]; MCP-1 (CCL2), which recruits monocytes [75,80]; and soluble intercellular adhesion molecule (sICAM)-1 [81], among others [69,70]. Additionally, the Akt (also known as protein kinase B) and mitogen-activated protein kinase (MEK)/extracellular-signal regulated kinase (ERK) intracellular signaling routes, together with signal transducer and activator of transcription (STAT) proteins and SMAD transcription factors have been described as key molecular pathways that play a role in the mechanisms leading to the development of endometriosis [82,83,84], as reviewed by Aznaurova et al. [85]; McKinnon et al. [86]; Riccio et al. [87]; Patel et al. [47]; and Zhang et al. [48].

## 4. Endometriosis and Diet

There is also a great interest in uncovering the impact of diet components on both endometriosis development and treatment, mostly based on growing evidence of their relationships with chronic inflammation, redox state, estrogen activity, prostaglandin metabolism, and the menstrual cycle, among others. Although epidemiological findings obtained from these studies are difficult to consistently replicate, recent reports and reviews suggest that fruits and vegetables, calcium and vitamin D from dairy products, and fish oils and Omega-3 polyunsaturated fatty acids are related to a lower risk of endometriosis [88,89]. On the contrary, a high intake of trans fats and red meat, processed or not, is highly associated with endometriosis development [88], while consumption of poultry, fish, shellfish, and eggs is not associated with this disease [90,91]. Recently, Nodler et al. [92], in a prospective cohort study, showed that greater dairy consumption, specifically yogurt and ice cream, during adolescence is associated with a lower risk of laparoscopically-confirmed endometriosis [92]. Regarding fruit and vegetable consumption and endometriosis risk, the recent work of Harris et al. [15] in a wide prospective cohort study reported an inverse association between greater fruit intake and risk of laparoscopically confirmed endometriosis, especially for citrus fruits (women with ≥ 1 servings of citrus fruits/day had a 22% lower endometriosis risk (95% CI = 0.69–0.89; *p* = 0.004) compared to those consuming < 1 serving/week. Furthermore, consumption of ≥ 1 servings/day of cruciferous vegetables (cauliflower, cabbage and Brussels sprouts, but not broccoli) was significantly associated with a higher endometriosis risk [15]. As Huijs and Nap [93] stated in their review, the effect of specific food products on endometriosis symptoms and whether they provide or not a synergic effect remains unclear.

The link between diet and endometriosis highlights the potential of the anti-inflammatory compounds present in foods to alleviate endometriosis symptoms and their progression. However, there are current gaps that preclude obtaining reproducible and clear interpretable results. First of all, these studies are based on retrospective or case-control populations and diet intake relays in self-reported food frequency questionnaires, which could lead to a bias whereby females report greater intake of those foods considered healthier. Additionally, contribution of the level of EDC food contamination, for example, through pesticides in fruits and vegetables, or estrogenic EDCs present in farmed animal foods, cannot be taken into account. All of these considerations underscore the necessity to perform new case-control studies in endometriosis patients eating strictly controlled diets (type of food, cooking and food preparation, and duration of the intervention) using foods rich in anti-inflammatory compounds by studying the level of endometriosis inflammatory markers and the clinical evolution.

## 5. Brassicaceae and Their Bioactive Compounds in Inflammation

As mentioned before, *Brassica* crops are well known for their high content in health-promoting compounds [94,95]. Specifically, glucosinolates (GLSs) and their bioactive breakdown product, ITCs, have shown different types of activity, such as induction of detoxification Phase II enzymes [96] and anti-tumorigenic [97] or anti-inflammatory effects [98]. Glucosinolates are secondary plant metabolites that are mainly present in vegetables from the Brassicaceae family. Their basic structure is a thiohydrozimate-O-sulfonate group linked with a glucose, whose side chain can vary depending on the amino acid from which they are derived [99]. Inside the plant cells, GLSs are stable, but when tissue disruption takes places, for example, chewing or wounding, these biomolecules are hydrolyzed by the enzyme myrosinase (EC 3.2.1.147), producing ITCs [100]. Cooking methods can also affect the content in glucosinolates and its degree of conversion to ITCs. For example, steaming, microwave cooking, and stir frying are processes that reduce the glucosinolate content the least and can even improve its extraction from the food matrix [101]. However, boiling or blanching usually inactivate myrosinase while leaching glucosinolates and breakdown products into the water [102]. A link between chronic inflammatory-related diseases, such as cancer or obesity, and these bioactive ITCs has been reported [10,103]. Furthermore, different ITCs are involved in diverse mechanisms and pathways of inflammatory processes; this analyzed here under the perspective of endometriosis.

### 5.1. Aliphatic ITCs and Related Metabolites

One of the most frequently studied aliphatic ITCs is sulforaphane (SFN), the resulting hydrolysis derivative of glucoraphanin, due to its effects on human health. SFN is widely considered to be a multi-faceted agent due to its role in several cellular pathways—attenuating, reversing, or even blocking different activities in the cellular metabolism [97]. For example, it has been reported that SFN specially induces the activation of Phase II detoxification enzymes, alters cellular signaling pathways, and takes part in the suppression of pro-inflammatory responses [104]. Specifically, SFN has been described as an inductor of the Nrf2 transcription factor (Figure 1A), which is responsible for the transcription of various genes involved in antioxidant activities or anti-inflammatory pathways [105]. The influence of SFN is based on its ability to bind the cysteine residues present in the Nrf2 repressor Keap1. under normal conditions, Keap1 promotes the degradation of Nrf2 in the proteasome, but when SFN interacts with it, Nrf2 is released and translocated to the nucleus [106].

Furthermore, SFN is involved in the NFκB pathway, decreasing its ability to bind to target genes related to the inflammatory response, such as pro-inflammatory interleukins, including tumor necrosis factor (TNF)-α [97]. In this signaling cascade, Toll-like receptor (TLR)-4 is the first element, and it is responsible for recognizing pathogen-associated molecular patterns (PAMPs) or intrinsic molecules (Figure 1B), such as fibronectin and heparan sulfate [107,108]. It has been reported that SFN can bind to the cysteine residues in this receptor, suppressing its oligomerization and inhibiting the inflammatory response [109].

Additionally, several pre-clinical studies have demonstrated the efficiency of this ITC in models of chronic inflammatory diseases [110]. For example, Zhao et al. [111] demonstrated that the administration of SFN to a nephropathy murine model reduced the level of reactive oxygen species (ROS) and activated Nrf2 transcription factor. However, little to no information about the effects in models of endometriosis has been found. Recently, Zhou et al. [112] revealed that SFN attenuated endometriosis symptoms in rats by diminishing the levels of TNF-α and IL-6 in peritoneal fluid and plasma. Moreover, SFN seems to decrease the expression of VEGF, affecting the neoangiogenesis of the endometriotic foci. In addition, SFN has been reported to inhibit the growth of ectopic endometrial tissue in sciatic endometriosis rat models, showing a decrease in both the size of lesions and the VEGF level [113]. However, more studies are needed in order to determine the effect of SFN in long-term dietary interventions.

SFN has a structural analog, sulforaphene (SFE), which differs only by having a double bond in the alkyl chain. It is derived from the aliphatic GLS glucoraphenin and can be found mainly in radishes [114]. Although research on SFE has been focused on its anti-carcinogenic and pro-apoptotic effects [115], little is known about its direct effect on inflammatory processes. However, it is known that SFE can lose its double bond, turning into SFN, whose properties are better known [116,117].

Another aliphatic ITC present in *Brassica* foods, including broccoli sprouts, is erucin [118]. Although an inter-conversion of SFN to erucin and vice versa has been reported in humans after its consumption, the effects of erucin in inflammation are also significant [119]. For example, it has been reported that it decreases the DNA-binding capacity of NFκB, thus reducing the transcription of target genes related to the inflammatory process, including TNF-α, IL-6, COX-2, and inducible nitric oxide synthase (iNOS) [120]. Additionally, glucoiberin is a glucosinolate that is present in products such as cabbage and kale, and its degradation product is iberin [121,122]. As mentioned previously, TLRs play an important role in the induction of the innate immune response, and iberin can prevent the dimerization of TLRs, down-regulating NFκB signaling [107,123].

In oilseed rape and other herbaceous *Brassicas,* the aliphatic ITC allyl-ITC (AITC) derived from sinigrin does not have sulfur atoms in its side chain, unlike SFN and SFE. The action of AITC in the NFκB pathway when administered to the LPS-activated macrophage RAW 264.7 has been reported to decrease the production of TNF-α, IL-6, and nitric oxide [124]. In addition, in vivo studies performed in rat models of mammary carcinogenesis showed that AITC administration did not provoke a decrease in p65-NFκB expression and, thus, in pro-inflammatory cytokines such as IL-6 [125]. The same group also reported that AITC has a preventive effect on DMBA-induced mammary carcinogenesis by modulating the aryl hydrocarbon receptor (AhR)/Nrf2 signaling pathway [126]. This cytoplasmic receptor and transcription factor has a major role in environmental pollutant detoxification [127]. Nevertheless, recent studies have highlighted the role of AhR as a negative regulator of the immune response [128]. In this regard, it has been reported that AhR-null mice produce higher levels of pro-inflammatory cytokines, such as TNF-α and IL-12 [129].

### 5.2. Indoles and Related Compounds

The hydrolysis of glucobrassicin, a major indole glucosinolate present in *Brassicas*, including broccoli or cabbage [130], produces indole-3-carbinol (I3C). I3C can be converted into its dimeric condensation product 3,3-diindolylmethane (3,3-DIM) [131]. Both biomolecules are AhR agonists that decrease the level of IL-1β and increase the rate of the detoxification cascade [132]. In addition, when I3C was administrated to murine models of doxorubicin-induced damage, positive stimulation of the Nrf2 response and down-regulation of p50-NFκB expression was observed [133]. However, when macrophages derived from monocytes of systemic lupus erythematosus patients were treated with I3C, a switch toward a M2 phenotype was described [134]. As mentioned before, M2 macrophages could help with the survival of endometrial cysts [54]. Thus, in the absence of in vivo studies of I3C applied to endometriosis, the specific effect on this disease remains unknown.

On the other hand, 3,3-DIM alone has been reported to inhibit the inflammatory response by also down-regulating the NFκB pathway and decreasing the levels of PGE_2_, TNF-α, IL-6, and IL-1β in murine macrophage cell cultures [135], and the oral administration of 3,3-DIM to a model of acute colitis in mice provoked the down-regulation of various types of VEGF and the expression of the VEGF receptor-2 [136]. This function would be beneficial for endometriosis patients to reduce the neoangiogenesis of lesions. In addition, the effects of 3,3-DIM have been also tested in combination with dienogest, a common progestrin that is commonly prescribed to treat pain associated with endometriosis. Women who were administered with this combination for 3 months reported less endometriosis-associated pelvic pain [137]. However, only eight patients finished the study; therefore, studies with a larger number of volunteers are needed to better understand its effects.

Ascorbigen is a biomolecule obtained from glucobrassicin degradation products generated after the enzymatic hydrolysis of I3C in the presence of ascorbic acid that has shown high antioxidant potential [138]. Fermented cabbage extracts enriched in ascorbigen were found to have improved antioxidant capacity and nitric oxide inhibition in RAW 264.7 murine cell cultures [139]. Nonetheless, information about the exact effects of this metabolite in inflammation is very limited since no recent in vivo studies are available.

## 6. Aquaporins in Endometriosis

Aquaporins (AQPs) belong to the membrane intrinsic protein (MIP) family [140] and are found in practically all living organisms [141]. AQPs are tetrameric, small (25–34 kDa), hydrophobic proteins that constitute integral membrane channels [142]. The main function of AQPs is water transport and flux water regulation in cellular membranes [143]. Since their discovery, AQPs have been the focus of many studies, given the importance of water transport in all biological processes, and have been shown to be involved in several diseases of different tissues [144]. Specifically, nine of the thirteen total AQPs (AQP1, 2, 3, 4, 5, 6, 8, 9, and 11) discovered in mammals have been identified as existing in the human female reproductive tract [145]. Proper water transport and homeostasis is crucial for normal reproductive performance and, thus, these AQPs play key roles in the female reproductive system, as they are involved in vaginal lubrication, cervical ripening, implantation, uterine water inhibition, blastocyst formation, ovum transport, follicle maturation, and oocyte cryopreservation [145]. However, apart from the main described functions, some aquaporins are also involved in biological alterations of the reproductive system such as cell migration, proliferation, carcinogenesis, and inflammatory processes [146,147]. As previously described, these processes are crucial for the development of endometriosis, and thus, the link between AQPs and the pathogenesis of endometriosis must be further explored [148]. AQP1 might have a role in endometriosis via the Wnt/β-catenin signaling pathway. Hence, AQP1 gene silencing entails a decrease in the adhesion and invasion of ectopic endometrial cells, as well as a decrease in angiogenesis and an increase in the apoptosis rate of these cells [149]. In the same way, AQP5, whose expression is menstrual cycle-dependent [150], is more highly expressed in eutopic as compared with ectopic endometrial cells in women with endometriosis [151]. Another investigation reported that AQP5 expression was induced by estrogen through activation of the estrogen-response element (ERE) in the promoter region of the AQP5 gene [152]. Therefore, it could be related to endometrial cell invasion and proliferation. This aspect was observed in a study where overexpression of AQP5 promoted endometrial tumor cell migration, while knockdown of AQP5 decreased migration of these cells [153]. Moreover, Choi et al. [154] described that AQP2 and AQP8 expression was significantly increased in the eutopic endometrial cells of patients with endometriosis as compared with a control group. Also, this work stated that the endometrial expression of AQP9 was significantly decreased in the eutopic endometrium of patients with endometriosis as compared with that of healthy women. Besides, it was demonstrated that the down-regulation of AQP9 increased cell migration and invasion through the activation of the ERK/p38 MAPK signaling pathways. These results are consistent with the demonstration of changes in the expression of AQP9 in several types of cancer and inflammatory diseases [155], which suggests that AQP9 could also play an important role in the pathogenesis of endometriosis.

Additionally, it has been reported that the expression of AQP1, present in macrophages, regulated their migration in a murine model of peritonitis [156], showing that functional AQP1 suppresses the migration of resting macrophages. Further, the expression of AQP1 was found to attenuate macrophage-mediated inflammatory responses in LPS-induced acute kidney injury by inhibiting ERK and p38 MAPK activation [157]. In addition, AQP3 has shown to be a key component in the immune function of macrophages through its participation in water and glycerol membrane transportation, as well as in their phagocytic and migration activity [147]. Furthermore, AQP4 blockade alleviated the development and severity of irradiated lung damage in a mouse model. This was associated with attenuated infiltration of inflammatory cells, decreased production of pro-inflammatory cytokines, and inhibited activation of M2 macrophages [158].

Therefore, the available results suggest that a relation between AQPs and endometriosis would not only be mediated by their role as water membrane channels (Figure 2), which is their major physiological function [143]. Their involvement in several intracellular signaling pathways as signal regulating factors (via transporting molecules or coupling with other proteins) [159] should be taken into account. Particularly, AQPs 1, 2, 8, and 9 could be related to endometriosis due to their participation in different signaling pathways such as Wnt/β-catenin, ERK/p38 MAPKm, and PI3K/Akt [149,152,154]. Hence, more studies are needed to reveal the actual signaling regulatory function of AQPs. In contrast, the roles of AQP1 and 5 in different inflammatory processes, whose regulation is altered in endometriosis (displaying maintained or increased expression), could be mediated by their roles in water fluid homeostasis as water transporters, and thus their potential relevance in endometriosis should be also further investigated [147,160].

### Modulation of Aquaporins by Bioactive Compounds: Further Targeted Therapies

The modulation of the presence or functionality of aquaporins by bioactive compounds or drugs may provide new opportunities for therapeutic applications in a variety of diseases that involve alterations of cellular homeostasis and inflammatory processes [161]. In this sense, gating of the AQP channel by different non-covalent small molecule inhibitors has been reported [162]. Blockers of AQP4 have demonstrated a positive effect in ischemia-mediated cerebral edema models [163]. Also, some of the quaternary ammonium compounds [164,165,166], arylsulfonamide-based carbonic anhydrase inhibitors [167,168], and certain anti-epileptics [169] have been suggested to be selective blockers of AQP1 and AQP4. Furthermore, on the other hand, there are pharmacological synergists to AQP functionality, i.e., the aryl sulfonamide-related derivatives [170] and the chemical derivative of the arylsulfonamide compound furosemide [166], that directly and specifically potentiate AQP-mediated water transport.

Aberrant expression of AQPs in eutopic and ectopic endometrial cells could be a target of action against endometriosis disease. Some studies of endometriosis cell models have reported a deficit in antioxidant enzymes and, thus, a low antioxidant capacity. Therefore, excess accumulation of reactive oxygen species could play a role in the infertility associated with endometriosis [171]. AQP8 has been reported as a fine regulator of redox signal transduction since it regulates permeability to water and H_2_O_2_ under stress conditions [172]. As described above, AQP8 is overexpressed in eutopic endometrial cells of women with endometriosis [154]. Hence, the regulation of AQP8 expression could reduce the level of ROS in cells, therefore reducing oxidative stress. This could favor better evolution of endometriosis. In this sense, a recent study carried out in the erythromegakaryocytic cell line B1647 showed a downregulation of AQP8 expression with low doses of SFN (10–30 µM) [173]. Therefore, SFN could be a good therapy to regulate AQP8 expression in endometrial cells and decrease oxidative stress in these cells.

Furthermore, if we consider that endometriosis is an estrogen-dependent disease [174], new therapies that counter the effect of estradiol need to be explored. Estrogens are steroid hormones that regulate AQP expression in the female reproductive system [175]. The expression of AQP2 and AQP5, which is up-regulated in endometrial cells and seem to play a role in the estrogen-related migration, invasion, adhesion, and proliferation of these cells, could be regulated to act on the ERE located in the promoters of the genes encoding AQP2 and AQP5 [152,176]. In some studies, SFN [177] and I3C [178,179,180] have been shown to be modulators of the estrogen effects in several types of cancer. Given the common features of cancer and endometriosis, these compounds could induce changes in the ERE of the AQP5 gene, thus regulating its expression. In the same way, it has been reported that sulforaphane protects the central nervous system in traumatic brain injury patients by reducing the loss of AQP4 in the injury core and increasing the level of AQP4 in the penumbra region [181,182].

All of the above data indicate that AQP activity can be modified through distinct mechanisms. However, it is also important to consider that pharmacological and bioactive natural compounds might interact regarding AQP expression and/or functionality, reflecting diverse affinity and efficacy. Therefore, it has become clear that the investigation of plant biomolecules as glucosinolates/isotiocyanates that target the modulation of AQPs is an interesting research objective, especially as these are natural compounds with fewer side effects than other therapeutic agents.

## 7. Conclusions and Perspectives

To summarize the results presented in this review, macrophages are key immune cells in endometriotic lesions, which are characterized by an anomalous inflammatory environment and cell growth in women with endometriosis. We have presented and discussed a new approach based on the potential role of glucosinolates/isothiocyanates from brassicas as putative natural anti-inflammatory compounds, which could reduce the symptoms and progression of endometriosis, improving the quality of life for affected women. The potential role of glucosinolates in the development chronic inflammatory diseases should be investigated since there is still little information in this respect. In connection to that, water transport and solute movement, in which AQPs are involved, are crucial events not only in the maintenance of immune homeostasis but also in human pathophysiology. The reported importance of AQPs in different inflammatory processes could be integrated in the role played by macrophages in endometriosis. Therefore, the modulation of cellular water and solute transport through AQPs by glucosinolates may provide new opportunities for the design of preventive and therapeutic approaches to improve the quality of life of women with endometriosis.

## 8. Materials and Methods

A non-systematic approach was chosen and a narrative synthesis of the results of the searched articles was carried out in accordance with Gasparian et al. [183] and Saracci et al. [184].

## Figures and Tables

**Figure 1 ijms-21-09397-f001:**
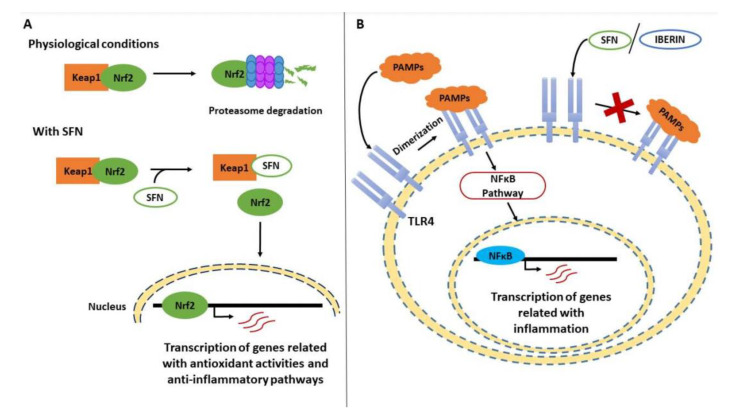
Effect of *Brassicaceae* bioactives compounds upon the transcription of genes related with inflammation. (**A**) In physiological conditions, Keap1 promotes the degradation of Nrf2 by the proteasome. However, when sulforaphane (SFN) interacts with Keap1 by binding its cysteine residues, Nef2 is released and translocated to the nucleus. In this way, Nrf2 up regulates the transcription of genes related with antioxidant activities and anti-inflammatory pathways. (**B**) In an inflammatory process, Toll-like receptor 4 (TLR4) undergoes dimerization after recognizing pathogen-associated molecular patterns (PAMPs), activating the NFκB pathway and up-regulating genes related with inflammation. When SFN and iberin interact with TLR4, it cannot oligomerizate, and the inflammatory response is inhibited.

**Figure 2 ijms-21-09397-f002:**
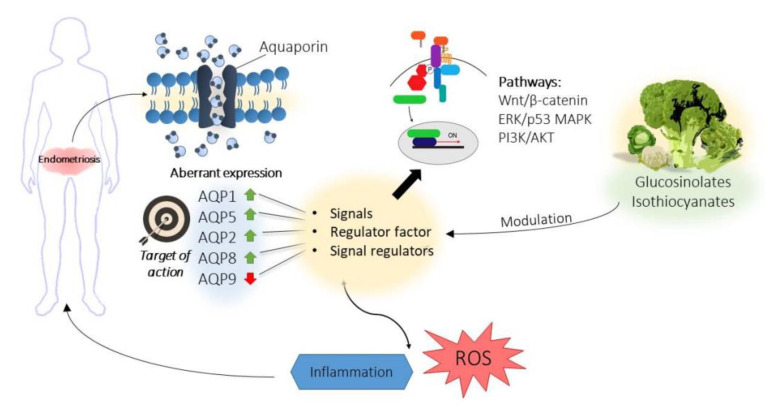
Role of aquaporins in endometriosis. Aquaporins are involved in different signaling pathways related with inflammation and endometriosis. Besides, aquaporins present an aberrant expression in endometrial tissues of women with endometriosis and therefore, they could be the target of therapies based on bioactive compounds derived from brassicas.

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
