# Peer review of "Brassica Bioactives Could Ameliorate the Chronic Inflammatory Condition of Endometriosis"

_ijms, 2020, doi:10.3390/ijms21249397_

Round 1

Reviewer 1 Report

It would have been interesting to study if the dietary compounds the paper talks about are influenced by heat(or other form of food processing)in order for the information to be more useful for patients suffering from endometriosis and clinicians offering dietary and lifestyle changing ideas to their patients. 

Author Response

It would have been interesting to study if the dietary compounds the paper talks about are influenced by heat (or other form of food processing) in order for the information to be more useful for patients suffering from endometriosis and clinicians offering dietary and lifestyle changing ideas to their patients. 

**Following Reviewer’s suggestion the requested information has been included in section 5: Brassicaceae and their bioactives compounds in inflammation (lines 249 to 254).

Reviewer 2 Report

ijms-1011584

The Manuscript entitled “Brassica bioactives could ameliorate the chronic inflammatory condition of Endometriosis”, deals with the role of Brassica crops as a potential option for improve the chronic inflammatory condition of endometriosis. Authors performed a review to search the literature in order to find out relevant articles and revealed the potential role of glucosinolates, bioactive derivates of Brassia crops, on chronic inflammatory disease.

The paper falls within the aim of International Journal of Molecular Sciences.

The topic is really attractive and I suggest it could be interesting for readers. The paper is well written and has useful figures. Nevertheless, authors should clarify some issues and improve the introduction citing relevant and novel key articles about the topic.

Authors should consider the following recommendations:

- Manuscript should be revised by a native English speaker in order to correct several typos.

- Abstract: Authors should include the conclusions of their review, particularly regarding the role of glucosinolates.

- Introduction: “Nowadays, it is estimated that approximately 176 million women are affected by endometriosis worldwide, which means about 10% of women of reproductive age.” Please add reference to this paragraph.

- Introduction: Endometriosis is a chronic inflammatory disease with a huge impact on health-care costs and, particularly, on women’ quality of life. Authors should underline this last one aspect, briefly referring to PMID: 31755667 and PMID: 31726815.

- Materials and methods: This section is missing. Authors should describe how did they perform this review. Authors could write that “opted for a non-systematic approach and performed a narrative synthesis of the results from searched articles”, (referring to PMID: 31553532; PMID: 21800117).

Author Response

The Manuscript entitled “Brassica bioactives could ameliorate the chronic inflammatory condition of Endometriosis”, deals with the role of Brassica crops as a potential option for improve the chronic inflammatory condition of endometriosis. Authors performed a review to search the literature in order to find out relevant articles and revealed the potential role of glucosinolates, bioactive derivates of Brassia crops, on chronic inflammatory disease.

The paper falls within the aim of International Journal of Molecular Sciences.

The topic is really attractive and I suggest it could be interesting for readers. The paper is well written and has useful figures. Nevertheless, authors should clarify some issues and improve the introduction citing relevant and novel key articles about the topic. Authors should consider the following recommendations:

- Manuscript should be revised by a native English speaker in order to correct several typos.

We asked the MDPI Editorial tools for English linguistic review to improve also the current version of the manuscript.

-Abstract: Authors should include the conclusions of their review, particularly regarding the role of glucosinolates.

Following Reviewer’s suggestion conclusions have been included in the abstract (lines 31-38).

- Introduction: “Nowadays, it is estimated that approximately 176 million women are affected by endometriosis worldwide, which means about 10% of women of reproductive age.” Please add reference to this paragraph.

Following Reviewer’s comment we have added the appropriated reference (Zondervan et al. 2018) (line 56):

Zondervan KT, Becker CM, Koga K, et al (2018) Endometriosis. Nat Rev Dis Prim 4:9. https://doi.org/10.1038/s41572-018-0008-5

- Introduction: Endometriosis is a chronic inflammatory disease with a huge impact on health-care costs and, particularly, on women’ quality of life. Authors should underline this last one aspect, briefly referring to PMID: 31755667(La Rosa et al. 2020) and PMID: 31726815 (la Rosa et al. 2020).

Suggested references have been included in the text (Lines 60-62 and Lines 64-66).

- Materials and methods: This section is missing. Authors should describe how did they perform this review. Authors could write that “opted for a non-systematic approach and performed a narrative synthesis of the results from searched articles”, (referring to PMID: 31553532; PMID: 21800117).

According to Reviewer’s suggestion a Material and methods section (Section 7) referring to PMID: 31553532 and PMID: 21800117 has been added before the Acknowledgements section (lines 519 - 522).

Reviewer 3 Report

This is an interesting narrative review about the role of brassica bioactives in the amelioration of chronic inflammation process which occurs in endometriosis. 

The authors have managed to analyse and describe extensively the pathophysiological pathways of inflammation which derive from endometriosis. 

I believe it should be accepted at its current form. 

Author Response

This is an interesting narrative review about the role of brassica bioactives in the amelioration of chronic inflammation process which occurs in endometriosis. The authors have managed to analyse and describe extensively the pathophysiological pathways of inflammation which derive from endometriosis. I believe it should be accepted at its current form. 

** Thank you very much for your positive evaluation and revision of the work. We tried also our best to improve the work with incorporation of the corrections/comments/changes suggested also for the rest of reviewers and editorial team.

Round 2

Reviewer 2 Report

Authors have performed the required changes, significantly improving the quality of the paper.